# OpenReview forum: "Let's Measure Information Step-by-Step: AI-Based Evaluation Beyond Vibes"
_TMLR — Accepted by TMLR_

### Review · Reviewer_9kgX · 2025-10-19

**Summary Of Contributions:**

The paper studies evaluation of texts without ground truth, with applications from peer reviews and open-ended summarization tasks. Given a pair of text, the evaluation mechanism rewards by the mutual information between the two. The main results include the following.

* Theoretical result.
  - The incentive analysis of the overseer is directly implied by previous work on peer prediction.
  - The **main results** is a sample-complexity bound on the estimation of f-divergences. The bound suggests that bounded f-divergences may have a better sample complexity than KL divergence used in previous work of GEM and GPPM.

* Empirical results. Following the main theoretical result, the paper empirically compares f-divergence based peer prediction mechanism with KL based mechanisms in previous work. The paper conducts experiments on peer reviews, summarization and translation texts. For each task, the paper measures discrimination between truthful report and problematic report (including strategic and low effort). The paper also studies the robustness of the mechanisms to semantic preserving attacks. The main findings suggest that peer prediction based mechanisms outperform LLM judges. Moreover, among peer prediction based mechanisms, TVD-MI outperforms KL based mechanism.

**Audience:**

Yes

**Audience Explanation:**

The paper compares mutual information (peer prediction) based mechanisms for benchmarking LLM evaluators without ground truth. LLM evaluation without ground truth, robustness to gaming are topics relevant for TMLR readers and widely important to the machine learning community.

**Broader Impact Concerns:**

The Broader Impact section is clear and sufficiently addresses foreseeable concerns. I have no additional concerns.

**Claims And Evidence:**

No

**Claims Explanation:**

As the main finding in the paper, both theoretical results and empirical results suggest that TVD-MI outperforms due to lower sample complexity than KL based mechanism. However, as the paper also points out, the experiments does not support the theoretical hypothesis of a linear degradation rate in sample size. The paper provides explanations in the appendix, suggesting a quadratic bias-variance trade off in the compression / effect size relationship. It is intuitive why this quadratic relationship explains a bias-variance trade off, but not made clear in the paper.

Confounding factors including writing style, length, etc. Previous work on GEM and GPPM suggests that evaluation performance for KL based mechanism is better with paraphrasing of both texts for comparison to remove confounding factors. The empirical experiments in the paper do not seem to follow the observation in previous work. Thus, it is unclear whether the empirical evaluation is fair enough to support and reflect the difference between the two mechanisms.

**Requested Changes:**

* Center the main contribution on the sample-complexity result. It would be more helpful for the readers to understand the contribution of Thm 3.3 and its implications in the empirical experiments if the paper de-emphasize the incentives for overseer. The incentives of the overseer is straightforward. When I read through the early sections in the paper, I thought main focus of the empirical evaluation is on the manipulations from the overseer.
* Tighten theory/empirical link. According to my comment above, it will be very helpful if the main text has a discussion of the main assumptions, formal claim, and a confirming plot on the compression / effect size relation.
* Replication of previous work. Also according to my comment above, it will be helpful if preprocessing can be added to remove confounding factors found in previous work.
* Terminology and citations.
  - Related work for peer prediction: "However, these methods separate evaluation into pre-processing and scoring, which confounds formal analysis of adversarial settings." I'm not sure how pre-processing matters to the evaluation. I thought they only help with removing styling confounding factors of texts.
  - Adversarial robustness. It would be helpful to use "sample complexity" instead of "adversarial robustness" in Thm 3.3., which is more standard in theory. Adversarial robustness typically means attacks from payoff-aware adversaries, instead of worst-case guarantee where worst-case is over possible joint distributions. Talking in the classical terminology, Elicitation GPT has an analysis of the expected payoff upper bound for an uninformed adversary. GEM/GPPM has an analysis of the epsilon IC under errors. Table 1 doesn't seem appropriate.

Minor suggestions:
* It would be helpful to add a roadmap to the paper.

---

### Review · Reviewer_PcM3 · 2025-11-03

**Summary Of Contributions:**

The paper proposes TVD-MI, a new mutual information-based framework that can be used to evaluate AI systems without ground truth. The key novelty of the method that distinguishes it from prior methods, as well as the LLM-as-a-judge framework, is the idea of prompting an LLM to assess whether two outputs are judging the same item or not, which is found to be a more robust measure than directly asking about the qualities. The authors provide theoretical guarantees of the method in terms of the error bound while estimating the mutual information from finite samples. Extensive experiments from various tasks, different measures, and comparisons with multiple baselines are provided.

The main strength is the extensive experiments. I believe this paper's experiments are more comprehensive than prior works like GPPM.

The main weakness is the writing. Many parts are confusing, especially the theory section and the implementation of the method.

**Audience:**

Yes

**Audience Explanation:**

The paper directly speaks to the evaluation of AI systems, which is an important problem for the general ML and AI community. The mutual information framework for evaluation has been investigated before, also in the context of ML.

**Broader Impact Concerns:**

A major question is related to fairness. If peer prediction is used for evaluation, it might provide a biased score: it seems that an agent with a minority opinion will be scored lower than the majority, no matter how much effort it exerts, because their opinions tend to be different from most of its peers. For example, in peer review, the majority of reviewers do not judge the correctness of the proofs. If a reviewer is particularly interested in verifying the proofs, he will be scored lower by the mechanism even than a low-effort reviewer who speaks more about the common issues like experiments. This is because it is very unlikely for him to be paired with a review that comments on the proof, so it's harder for an LLM to observe that they are judging the same paper. Will this be an issue for the proposed method, and if so, can the authors justify, maybe by highlighting the conditions under which the method can be used?

**Claims And Evidence:**

No

**Claims Explanation:**

Overall, the main claim of the paper --- the proposed TVD-MI is a more robust evaluation metric than prior work and the LLM judge framework --- is properly supported by the experiments. However, I have a few questions about the implementation of the method, which may affect my understanding of the contribution.

**Requested Changes:**

* First of all, I believe that the paper requires some non-trivial rewriting. This is a necessary condition for acceptance. See more details below:
  * First, there should be a section that describes the implementation of the method in more detail. A particular confusion of mine is how the mechanism computes the (lower bound) of MI using a binary output from an LLM judging whether two answers are from the same source.
  * It's not clear how the theory section helps the claim of the paper. It seems to show that if one can obtain some iid sample from the joint distribution and the products of marginals, then a lower bound of MI can be approximately estimated. Isn't it already known in prior work? I think this theory justifies the use of MI, which is not new, but not the idea of inferring whether two answers are from the same source, which is new. The authors may consider clarifying the interpretation of their theoretical results.
  * Related to the above points, the theory section needs some significant rewriting. For example, it's not clear how Def 3.2 is used.
  * What is the prompt for the LLM judge with context in different tasks (review, summarization, translations)? Is ref vs on ref indicating with context vs without context?

* Overall, the experiments are solid. However, I have a few comments about clarifications and suggestions for a few additional experiments.
  * A major concern about the experiments is that the AUC for baselines is smaller than 0.5 for ICLR tasks. This seems problematic as even an uninformative random guess should result in an AUC of 0.5. The authors say the reason is the high compression ratio. However, I don't see how any meaningful evaluation methods can be fooled by such simple manipulations. I suggest the authors can pay more attention to this and provide a more detailed justification. Or this might be a confusion of mine due to a misunderstanding of the method.
  * Although the theory guarantees that the method is not gamable, there are many practical issues that may result in a big theory-application gap. For example, if the method is based on judging whether two responses are of the same task or not, is it possible to game the mechanism by prompting an LLM to manipulate the content while guaranteeing that it will provide enough signals to indicate the task? This strategy targeting the proposed method seems to be missing from the manipulation pool.

---

### Review · Reviewer_Nuzg · 2025-12-21

**Summary Of Contributions:**

**Summary:**

This paper introduces a novel measure of mutual information (MI) between texts, tailored to textual data and addresses limitations of existing similarity and MI-based approaches. The authors provide a theoretical analysis of the measure’s properties and demonstrate its practical utility through extensive experiments across multiple text analysis tasks. Empirical results show that the proposed method consistently outperforms strong baselines, indicating its effectiveness in quantifying informational relationships between texts.

**Strengths:**

The paper proposes a novel measure of mutual information between texts and conducts extensive experiments to demonstrate its effectiveness.

**Weaknesses:**

However, several issues remain.

- **For Theorem 3.3:** The idea of using f-divergence as an alternative to Shannon mutual information in order to reduce sample complexity has been explored in prior work. For example, Schoenebeck and Yu proposed an approach for estimating f-divergence and established an upper bound on the sample complexity in “Learning and Strongly Truthful Multi-Task Peer Prediction: A Variational Approach.” I did not quite understand why establishing a (tight) lower bound is helpful here. If the purpose is to show that f-divergence is a better choice, an upper bound is needed here, that is, we need an algorithm that estimates f-divergence within polynomial sample complexity, but not only a lower bound.

- **For TVD-MI:** The model and the definition of TVD-MI are difficult to follow, and the exposition would benefit from substantial clarification.
    1. The definition of the overseer’s type is confusing.  What does category C represent? Why should the overseer have a type?
    2. What is TV(\cdot) in the definition of TVD-MI in Equation (3)? The notation was not introduced previously in the paper.
    3. It is not clear how CoT is used to to compute TVD-MI. What is A and how is A chosen?

- **For experiments:** Based on the examples in the appendix, I assume that the proposed method basically generate scores by prompting LLM judges, and the results mainly demonstrate that LLM judges are able to detect changes in mutual information. However, the results in Table 2-4 and Table 6 do not seem to effectively demonstrate the advantage of the proposed TVD-MI. Four methods appear to all effective for detecting strategic manipulation.

**Additional Comments:**

None.

**Audience:**

Yes

**Audience Explanation:**

Yes. The paper addresses a broadly relevant problem in ML—evaluating AI systems without ground truth—using an information-theoretic approach with both theoretical guarantees and empirical validation, which should be of interest to TMLR’s audience.

**Broader Impact Concerns:**

None.

**Claims And Evidence:**

Yes

**Claims Explanation:**

All the experiment results are supported with sufficient details. The main theorems are proved.

**Requested Changes:**

- The exposition needs significant improvements. The following questions should be clearly explained.
    1. The definition of the overseer’s type is confusing.  What does category C represent? Why should the overseer have a type?
    2. What is TV(\cdot) in the definition of TVD-MI in Equation (3)? The notation was not introduced previously in the paper.
    3. It is not clear how CoT is used to to compute TVD-MI. What is A and how is A chosen?
- It would be beneficial to strengthen the experimental results by demonstrating a larger advantage of the proposed method over prior approaches. As currently presented, all methods appear capable of detecting strategic manipulation, which might suggest that the generated data are too easy to distinguish.

---

> ### Author Response · Authors · 2025-12-21
> **Author Response to Nuzg**
>
> Thank you for the careful review and for the concrete requests. Your main concerns were about (1) how Theorem 3.3 should be interpreted relative to prior work on $f$-divergence estimation, (2) clarity of the TVD-MI definition and notation (overseer “type,” what TVD denotes, what (A) is, and how CoT is used), and (3) whether the experiments demonstrate a clear advantage of TVD-MI over prior approaches. Below we summarize clarifications made in the revised manuscript.
>
> **(1) Interpretation of Theorem 3.3 (why a lower bound / relation to prior work)**
>
> You noted that prior work studies estimating $f$-divergences with polynomial sample complexity, and asked why a lower bound is helpful rather than an upper bound with an estimator. **Theorem 3.3 instead bounds the output of any valid $(1-\delta)$ lower confidence bound (including our LLM-based TVD-MI implementation), not the accuracy of a point estimator.** It characterizes how much information can be reliably certified in the presence of adversarial ambiguity.
>
> **Edits We Have Made.** We clarified that Theorem 3.3 is not proposing a new estimator upper bound, but a *distribution-free certification ceiling* (an impossibility bound) for any $(1-\delta)$ **lower confidence bound** that depends only on the empirical type. We also added an explicit connection to prior $f$-divergence estimation results (including Schoenebeck & Yu), explaining that those results are complementary (algorithmic upper bounds under modeling assumptions), whereas our theorem isolates the worst-case barrier with an adversary that picks the joint distribution.
>
> ```tex
> \paragraph{Why a ceiling bound (and how it relates to prior $f$-divergence estimation).}
> Theorem~\ref{thm:wclb} is a \emph{distribution-free certification ceiling}: it upper-bounds the largest value that any $(1-\delta)$ \emph{lower confidence bound} can safely certify from $N$ samples when the estimator depends only on the empirical type. Prior work (e.g., \citet{schoenebeck2020learning}) studies \emph{estimation} sample-complexity of $f$-divergences, but assumes a functional class that contains an efficient or near-optimal scoring rule. Our result does not fix a functional class and instead considers a worst-case adversary that selects joint laws that are indistinguishable at the level of empirical types at finite $N$. Because we study certified lower bounds, Theorem~\ref{thm:wclb} characterizes the largest reliable value that any such estimator could hope to guarantee in this setting.
>
> ```
>
> **(2) TVD-MI definition and notation (overseer “type,” C, A, and TV)**
>
> You asked: what does category $\mathcal{C}$ represent, why does the overseer have a “type,” what is $\mathrm{TV}(\cdot)$, and how CoT is used to compute TVD-MI (what is $A$, how is it chosen?).
>
> **Edits We Have Made.** We clarified the exposition by defining notation writing TVD-MI directly as a standard hypothesis test lower bound on total variation distance. We also explicitly define total variation distance and make the acceptance set concrete (the “same source” label). Finally, we clarify that CoT is just the implementation of the test $r$ (a fixed prompting policy); only the final discrete label is used in the statistic.
>
> **(2.1) Clarifying the overseer as an agent in the game**
>
> ```tex
> \paragraph{The Overseer as an Agent.}
> The overseer is a computational procedure that observes a finite sample of paired reports and outputs a score. We emphasize two objects: (i) the overseer’s \emph{private signal}, which we take to be the empirical type, and (ii) a fixed \emph{decision rule} $r$ chosen before seeing the realized sample. Section~\ref{sec:certification} instantiates $r$ using an LLM prompting policy (Figure \ref{fig:latent-knowledge} (left)); the resulting test statistics yield certified lower bounds on $f$-mutual information.
> ```
>
> **(2.2) Notation fix - removed single-use C/A notation in Section 3.4**
>
> **(2.3) Clarify “CoT” paragraph in Section 3.4**
>
> **(3) Experiments: clarifying the *advantage* of TVD-MI over prior approaches**
>
> You wrote that the results “do not seem to effectively demonstrate the advantage of TVD-MI” and that “all methods appear capable of detecting strategic manipulation,” potentially suggesting the data are too easy. Here we note that TVD-MI vs Judge uses the same model while the Judge (ref) condition has additional context and we aggregate scores when computing Judge AUC scores. Despite these additional advantages, after controlling for false discovery, TVD-MI wins on AUC in 9/10 domains and remains above 0.70 under attacks, while the judge can fall near chance.
>
> **Edits We Have Made.** We made the comparison target explicit in Section 4.2 "Evaluation Mechanisms": TVD-MI’s advantage is (i) adversarial robustness, and (ii) achieving this while remaining black-box and reference-free (single-call critic) using the same base model as the judge.

---

> > ### Comment · Reviewer_Nuzg · 2026-01-07
> >
> > Thank you for the explanations.
> >
> > > We clarified that Theorem 3.3 is not proposing a new estimator upper bound, but a distribution-free certification ceiling (an impossibility bound) for any $(1-\delta)$ lower confidence bound that depends only on the empirical type. We also added an explicit connection to prior $f$-divergence estimation results (including Schoenebeck & Yu), explaining that those results are complementary (algorithmic upper bounds under modeling assumptions), whereas our theorem isolates the worst-case barrier with an adversary that picks the joint distribution.
> >
> > Thank you for the clarification. I fully appreciate Theorem 3.3 as an impossibility result and agree it is a strong theoretical contribution on its own. But my question is more about the implication of this impossibility result and how it supports the following sections of the paper. If I understood correctly, Theorem 3.3 demonstrates that $f$-divergence faces a less restrictive sample complexity barrier than Shannon Mutual Information. While this proves that $f$-divergence has the **potential** to be more tractable, it does not automatically imply that it is easier to estimate. To support the empirical results, a sample complexity upper bound for TVD-MI would be more effective.

---

> > > ### Author Response · Authors · 2026-01-07
> > > **Author Response to Nuzg**
> > >
> > > Thank you for the clarification. We agree that Theorem 3.3 is an impossibility result and does not by itself imply that $f$-divergences are easier to estimate as point estimators. Its role is different: it characterizes the limits of distribution-free certification.
> > >
> > > To clarify how this supports the rest of the paper, we emphasize the distinction between *certification ceilings* and *estimator-specific concentration*.
> > >
> > > First, Theorem 3.3 provides a distribution-free *certification ceiling*: it upper-bounds the value that any $(1-\delta)$ lower confidence bound can safely certify from $N$ samples under worst-case. Plugging in standard generators shows a qualitative separation in the amount of information that can be certified distribution-freely. For bounded $f$ (e.g., TV), the ceiling approaches the maximum possible information up to vanishing error, whereas for unbounded $f$ (e.g., Shannon/KL) the ceiling grows only logarithmically in $N$. As a result, an adversary can make true Shannon MI arbitrarily large while the *certifiable fraction* (and thus any payment based on certified MI) becomes arbitrarily small. This failure mode does not occur for bounded objectives.
> > >
> > > Second, separately from the certification ceiling, once a binary critic/test is fixed, our TVD-MI implementation estimates a *bounded statistic* (the difference between acceptance rates on real vs. shuffled pairs). Such bounded statistics admit standard $O(1/\sqrt N)$ concentration uniformly over distributions, once the critic is fixed. This is the sense in which the implemented estimator is tractable in practice. Theorem 3.3 does not address this estimator-specific behavior; instead, it explains why TVD is an appropriate choice when one requires distribution-free certification.
> > >
> > > We have made this separation explicit in the revised text, as summarized in our author update i.e. “Official Comment by Authors.”

---

### Author Response · Authors · 2026-01-07

We made two targeted clarifications in Section 3.3, *Statistical Limits for Gaming-Resistance*, to address questions raised by multiple reviewers.

**(1) Certification ceiling vs. estimator-specific concentration.**

We clarified the distinction between estimator-specific concentration for a fixed critic and the estimator-independent distribution-free certification limit captured by Theorem 3.3 in the introduction paragraph:

```latex
In this game, the overseer estimates $I_f(X;Y)$ from finite samples. Without knowledge of the response distribution, any estimator faces a worst-case adversary who can manipulate the distribution to minimize information while remaining consistent with observed samples. Our main robustness result, Theorem~\ref{thm:wclb}, upper-bounds the largest reliable lower bound any estimator can achieve. This yields a distribution-free sample-complexity bound under adversarial manipulation, extending McAllester and Stratos~(2020)’s indistinguishability construction from Shannon mutual information to general $f$-divergences. In the worst case, bounded, piecewise-linear $f$ (e.g., total variation) permit certification ceilings growing polynomially with sample size, whereas unbounded, super-linear $f$ (e.g., Kullback--Leibler) show only logarithmic growth, requiring exponentially more samples to certify extra nats. This separation motivates TVD-MI, estimated with a binary “real or shuffled pair?” critic (Fig.~\ref{fig:latent-knowledge}, left). For a fixed critic, a bounded test statistic concentrates as $O(1/\sqrt{N})$ \citep{BoucheronLugosiMassart2013}; Theorem~\ref{thm:wclb} instead characterizes the estimator-independent certification limit.
```

**(2) Practical implication: relative robustness of certified payments.**

We clarified the discussion of why a ceiling bound is the relevant object in our setting and how it differs from prior work on $f$-divergence estimation in the second section paragraph:
```latex
\paragraph{Why a ceiling bound (and how it relates to prior $f$-divergence estimation).}
Theorem~\ref{thm:wclb} is a \emph{distribution-free certification ceiling}: it upper-bounds the largest value that any $(1-\delta)$ \emph{lower confidence bound} can safely certify from $N$ samples when the estimator depends only on the empirical type. Prior work (e.g., \citet{schoenebeck2020learning}) studies \emph{estimation} sample-complexity of $f$-divergences, but assumes a functional class that contains an efficient or near-optimal scoring rule. Our result instead considers a worst-case adversary that selects joint laws indistinguishable at the level of empirical types at finite $N$, and therefore bounds how much information can ever be \emph{reliably certified}. For bounded $f$ (e.g., TV), this ceiling approaches the maximum possible information up to vanishing error, whereas for unbounded $f$ (e.g., Shannon/KL) it grows only logarithmically. Consequently, the amount of information that can be reliably certified, and therefore used as the basis for the mechanism evaluation, can be arbitrarily small relative to the true MI.
```

Together, these clarifications make explicit (i) the practical mechanism-level implication of Theorem 3.3 in terms of relative robustness of certified payments, and (ii) how our TVD-MI estimator achieves standard $O(1/\sqrt{N})$ concentration once the critic is fixed, while the theorem remains an estimator-independent distribution-free limit.

**New Reference**

Stéphane Boucheron, Gábor Lugosi, and Pascal Massart. Concentration Inequalities: A Nonasymptotic Theory of Independence. Oxford University Press, Oxford, 2013. ISBN 9780199535255.

---

### Decision · Action_Editor_5eAD · 2026-04-14

**Recommendation:** Accept with minor revision

**Additional Comments:**

Here is a summary of the reviewers' final takes on the paper:
    Two out of three reviewers mentioned that the experiments part of the paper is still somewhat weak. Specifically, one reviewer mentioned that the results in Tables 2--4 and Table 6 do not seem to sufficiently demonstrate the advantage of the proposed TVD-MI: four methods appear to all effective for detecting strategic manipulation.
    Both these reviewers also have continual concerns about the exposition of the paper.

I got detailed feedback from one reviewer in particular. My proposal is to accept this paper subject to moderate revisions, where officially speaking, I have selected the option "Accept with minor revision" (though I stress that the below comments will take a bit more than minor revisions to address).

The below is a direct quotation from the aforementioned reviewer.

"I re-read the paper and the author's response. Overall, the authors' committed modifications do help clarify the theory section. My current understanding is that their main theorem connects the sample complexity of different f-mutual information with their ability to extract information. Based on Theorem 3.3, they thus suggest TVD mutual information is the most robust estimator."

"I currently have two questions that I hope the authors can further clarify:

## Main concern (quoting reviewer)

The paper focuses more on adversarial robustness rather than the more fundamental rule of an evaluator --- selecting good (e.g., human-preferred, or accurate) responses. The argument is that these objectives are the same (Section 3.2) under assumptions. However, these assumptions (e.g., manipulation can only apply on the signal, or joint distribution between text can be accurately estimated) do not tightly hold in practice. For example, did the authors try the manipulation strategy, e.g., in peer review experiments, "Please construct your review to include as much paper-specific information as possible so that it closely aligns with another reviewer of the same paper"? This seems to be a plausible manipulation that degrades quality (because the review will be longer and the key information is diluted), while it might help game the proposed mechanism.

### Expected actions (quoting reviewer)

Did the author implement a targeted manipulation like this in their experiments? If not, is there a justification, particularly why the proposed metric is a good estimate for quality?

## Secondary question (quoting reviewer)

I hope the authors could better connect their experiments with Theorem 3.3 (main theoretical result), by explicitly verifying the KL vs TVD mutual information for the proposed mechanism. That is, use the same $P^+$ and $P^-$ estimated in TVD-MI, putting it through the KL divergence formula to get the KL-MI. Will this result in a worse performance as predicted by the theorem?

## Action Editor's summary regarding experiments

The reviewer mentions that the authors may need to do additional experiments. Alternatively, the authors must provide a strong justification for not doing additional experiments. Please note that I may ask the reviewer for feedback on the "minor" revision, so please try to fundamentally address their concerns.

**Audience:**

Yes

**Audience Explanation:**

The topic itself is very interesting, at the intersection of algorithmic game theory and machine learning (in particular, LLMs). Peer prediction and LLMs in particular continue to be very interesting topics to many (to put it lightly). I believe there definitely is a good fraction of the community that would be interested in these results.

**Claims And Evidence:**

Yes

**Claims Explanation:**

The answer is actually in between yes and no. From the experiments side, there is a continual critique (explained in more detail under "Additional comments". On the positive side, all reviewers are positive about the theoretical results themselves.

---

> ### Author Response · Authors · 2026-04-29
> **Summary of Camera-Ready Revisions**
>
> Thank you for the constructive feedback and decision to accept with minor revisions. We have revised the paper to clarify the scope of the theoretical result and guarantees, the limitations of the experiments, and the connection between the experiments and Theorem 3.3. We also added appendix comparisons (with discussion in main text) of mechanism scores across conditions for CNN/DailyMail and ICLR, including human gold reference responses, to make discussion of quality evaluation more concrete.
>
> **On the suggested manipulation.** The reviewer asked whether we tested a manipulation such as:
>
> > "Please construct your review to include as much paper-specific information as possible so that it closely aligns with another reviewer of the same paper."
>
> We did not test this exact adaptive prompt. We now state this explicitly and treat it as an important adaptive-overlap attack outside the scope of the current experiments. We have also replaced claims about “item-level quality” with “item-level detection” where appropriate.
>
> Our existing experiments test related, but non-adaptive, components of this concern: increasing paper-specific detail and injecting shared/spurious overlap. Responses in the peer review task follow the ICLR review template, and the high-paper-specificity behavior encouraged by the prompt is already close to strategies such as the Thorough Evaluation baseline, whose prompt includes, "Take meticulous notes covering all aspects of the paper including theoretical foundations, experimental methodology, results, and implications." This is a faithful high-detail strategy that scores highly. We test adding spurious shared content in the tampering experiments in Section 5.3. Those experiments provide evidence that TVD-MI is more robust than alternative methods under non-adaptive shared-overlap manipulations.
>
> **Connection to Theorem 3.3.** We have strengthened the discussion of the gaming/tampering experiments in Section 5.3 to make it clear they test the robustness claim of Theorem 3.3. Theorem 3.3 quantifies how many samples are needed to reliably certify that the mutual information exceeds some bound for a worst-case / adversarial distribution. It shows that bounded divergences such as TVD admit qualitatively stronger worst-case guarantees under adversarial manipulation than unbounded divergences such as KL. Our tampering experiments (Section 5.3 - Tables 5 and 6) were empirical stress tests inspired by Theorem 3.3 to test if this comparative robustness prediction appears by inserting overlapping content across responses.
>
> **On why the metric is still meaningful.** Our claim is not that information preservation is identical to quality in every setting; review quality itself is noisy and latent. Broadly, it is possible to improve a strategy by adding paper-specific overlap. The claim in Section 5.2 is narrower: the mechanism can separate faithful from problematic strategies as measured by effect-size and AUC. This positions TVD-MI as a robustness-oriented mutual-evaluation signal, but not a substitute for task-specific quality metrics. In Section 5.3 the claim is that attacks that work by injecting spurious shared content are limited under TVD-MI. We have revised Section 3.2 and the limitations to make this scope explicit. The DPI gives the cleanest guarantee, but does not rule out all adaptive strategies. In the main-text discussion, we now highlight examples that exhibit qualitative inversion: on CNN/DailyMail, fabricated “Conspiracy Theory” summaries receive higher judge scores than human reference summaries, and on ICLR, “Method Shift” responses can receive higher judge-without-context scores than reference reviews.
>
> **On the reviewer’s KL suggestion.** We agree that a KL-on-critic-output ablation is a useful diagnostic, but we clarify that it tests a different object from the KL-vs-TVD separation in Theorem 3.3. Our TVD-MI estimator is discriminative: it uses a critic to distinguish paired from shuffled responses, rather than explicitly estimating $P^+$ and $P^-$. One can compute KL after the low-cardinality critic representation, but the theorem concerns distribution-free certification of the underlying text-response distribution, whereas this ablation concerns the critic-induced representation. We have added this distinction explicitly.
>
> **Summary of Revisions.** Throughout the camera-ready, we improve sign-posting around the experimental scope of the theorem and limitations, added judgement comparisons with gold human references in the appendix with citation in the discussion section, replaced terms like “item-level quality” with “item-level detection” to emphasize that the mechanism does not directly measure quality, and use the term “mutual evaluation” consistently to describe the mechanism.